# Prevalence of Diabetic Retinopathy and Use of Common Oral Hypoglycemic Agents Increase the Risk of Diabetic Nephropathy—A Cross-Sectional Study in Patients with Type 2 Diabetes

**DOI:** 10.3390/ijerph20054623

**Published:** 2023-03-06

**Authors:** Wei-Ming Luo, Jing-Yang Su, Tong Xu, Zhong-Ze Fang

**Affiliations:** Department of Toxicology and Sanitary Chemistry, School of Public Health, Tianjin Medical University, Tianjin 300070, China

**Keywords:** diabetic nephropathy, diabetic retinopathy, oral hypoglycemic agents

## Abstract

Objective: This study investigated the effect of amino acid metabolism on the risk of diabetic nephropathy under different conditions of the diabetic retinopathy, and the use of different oral hypoglycemic agents. Methods: This study retrieved 1031 patients with type 2 diabetes from the First Affiliated Hospital of Liaoning Medical University in Jinzhou, which is located in Liaoning Province, China. We conducted a spearman correlation study between diabetic retinopathy and amino acids that have an impact on the prevalence of diabetic nephropathy. Logistic regression was used to analyze the changes of amino acid metabolism in different diabetic retinopathy conditions. Finally, the additive interaction between different drugs and diabetic retinopathy was explored. Results: It is showed that the protective effect of some amino acids on the risk of developing diabetic nephropathy is masked in diabetic retinopathy. Additionally, the additive effect of the combination of different drugs on the risk of diabetic nephropathy was greater than that of any one drug alone. Conclusions: We found that diabetic retinopathy patients have a higher risk of developing diabetic nephropathy than the general type 2 diabetes population. Additionally, the use of oral hypoglycemic agents can also increase the risk of diabetic nephropathy.

## 1. Introduction

Diabetes is a metabolic disorder caused by absolute or relative insufficient secretion of insulin, with type 2 diabetes (T2D) being the most common. In 2021, 537 million adults aged 20–79 years had diabetes around the world, and by 2045, the number of diabetic adults is expected to rise to 783 million [1]. T2D is associated with many adverse complications, and its complications are a major cause of death for T2D. Diabetic microangiopathy is a group of common complications of diabetes, in which diabetic retinopathy (DR) and diabetic nephropathy (DN) are the most common. DN is proteinuria and a progressive decrease in glomerular filtration rate (GFR) due to prolonged diabetes. The incidence of DN is also on the rise in China, and it has become the second cause of end-stage renal disease, second only to various glomerulonephritis. As both belong to microvascular disease, the correlation between DR and the risk of DN incidence has become a research topic. According to an UK study, DR occurs earlier than other complications [2]. The presence of DR means not only vision problems, but also an increased risk of other microvascular and macrovascular complications [3]. At present, some studies have found that DR occurs earlier than DN, can promote the development of DN, and can also help diagnose DN [4]. Additionally, DN patients with concurrent DR have an increased risk of rapid renal disease progression and generally worse renal outcomes [5]. Additionally, with the further application of metabolomics in the study of the pathogenesis of T2D and its complications, researchers have found that amino acid metabolism is closely related to microangiopathy. Amino acids, such as leucine (Leu) [6], histidine (His) [7,8], phenylalanine (Phe), and tyrosine (Tyr) [9], have been confirmed to be significantly related to the occurrence and development of DN in past studies. Metabolomics has gradually become an important method system to reveal the risk factors of DN. However, it is still unclear whether the relationship between amino acids and DN exists in people with different DR Status.

Metformin, Acarbose, and Sulfonylureas are all common oral hypoglycemic agents during the treatment of diabetes. At present, the protective effect of Metformin on the kidney has been widely studied. A study found that Metformin has a potential protective effect on DN through the AMPK/SIRT1-FoxO1 pathway [10], while other scholars thought there is no relation between them. However, less research has been done on the effects on DN risk of the other two drugs at the same time. Whether there are interactions between different drugs remains unclear. Thus, it is time to identify risk factors and early prediction of diabetes and its complications, which is critical for reducing diabetes complications [11,12] and economic burden [13]. Additionally, this is beneficial from both clinical and public health perspective [14]. It is against this background that we conducted the research on the effect of DR and the use of drugs on the risk of DN.

## 2. Materials and Methods

### 2.1. Study Method and Population

All the information of T2D patients is retrieved from the First Affiliated Hospital of Liaoning Medical University (FAHLMU), which is a tertiary general hospital located in Jinzhou, Liaoning Province, China. Inclusion criteria for this study were: (1) Patients diagnosed as T2D or treated with anti-hyperglycemic therapy; (2) The information of diabetic microvascular disease including DN and DR is completed. (3) The information on the use of Metformin, Acarbose, and Sulfonylureas drugs is completed. Exclusion criteria were: (1) T2D patients under the age of 18; (2) Subjects lacking amino acid indicators, height, weight, or blood pressure. (3) Patients with extreme outliers of amino acids. A total of 1821 patients with T2D were preliminarily included in this study. According to the exclusion criteria, 1031 subjects were finally included in this study, including 188 patients in the DN group and 843 T2D patients in the control group (Figure 1). Then, we used multiple imputation to deal with the missing data. 

The ethics of the study was approved by the Ethics Committee for Clinical Research of FAHLMU. Additionally, due to the retrospective nature of the study, informed consent was waivered, which is consistent with the Declaration of Helsinki.

### 2.2. Data Collection and Clinical Definitions

We retrieved the data from electronic medical records which included demographic and anthropometric information, and current clinical factors and information of diabetic complications. Demographic data included gender and age. Anthropometric measurements included weight, height, systolic blood pressure (SBP), and diastolic blood pressure (DBP). Clinical parameters included total cholesterol (TC), triglycerides (TG), glycosylated hemoglobin (HbA1c), low density lipoprotein cholesterol (LDL-C), high density lipoprotein cholesterol (HDL-C), creatinine (Crea), and uric acid (UA). Additionally, the duration of DN was recorded to exclude the interference of the duration of the disease on the results.

The measurements of anthropometric indicators were measured by standardized procedures in the hospital. Participants were allowed to wear light clothes and no shoes. weight and height were measured to the nearest 0.1 kg and 0.5 cm, respectively. Blood pressure in adults was measured with a standard mercury sphygmomanometer after a cuff on the right arm and after rest of 10 min in a seated position at an appropriate size. Age was calculated in years from the date of birth to the date of medical examination or hospitalization. The body mass index (BMI) was obtained according to the formula as the ratio of body weight (kg) to squared height (m) and classified according to the overweight and obesity criteria recommended by the National Health Commission of China [15]. The diagnosis and classification criteria of T2D in the study were based on the publishment of World Health Organization (WHO) or treated with antihyperglycemic therapy [16]. DR diagnostic standard based on eye exam results for T2D [17]. The diagnostic standard for DN was based on the criteria of care for T2D [18]. According to the RCS curve, His, tryptophan (Trp), Valine (Val), and threonine (Thr) were stratified according to 51 μmol/L, 46 μmol/L, 133 μmol/L, and 24 μmol/L, respectively (Figure 2). 

### 2.3. Amino Acid Quantification and Equipment

Details of the metabolomics assessment method have already been published [19]. Briefly, 8 h of fasting blood sample was collected at admission. A total of 22 amino acids were detected via LC-MS, i.e., asparagine (Asn), alanine (Ala), arginine (Arg), citrulline (Cit), Leu, lysine (Lys), Trp, Tyr, Thr, Val, glycine (Gly), proline (Pro), Phe, glutamine (Gln), His, methionine (Met), serine (Ser), ornithine (Orn), glutamate (Glu), aspartate (Asp), piperamide (Pip), and cysteine (Cys). AB Sciex 4000 QTrap system (AB Sciex, Framingham, MA, USA) was used to conduct direct injection MS metabolomic analysis. Analyst v1.6.0 software (AB Sciex) was used for data collection. ChemoView 2.0.2 (AB Sciex) was used for data preprocessing. Isotope-labeled internal standard samples were purchased from Cambridge Isotope Laboratories (Tewksbury, MA, USA). Standard samples of the amino acids were purchased from Chrom Systems (Grafelfing, Germany). 

### 2.4. Statistical Analysis

Continuous data were expressed as mean ± standard deviation (SD), non-normally distributed data were expressed as median (interquartile range), and categorical variables were expressed as numbers (percentages). In two populations with different DR prevalence status, it was tested whether there were differences between the different indicators of the patients in the DN group and the non-DN group. Continuous variables were normally distributed with *t*-test or ANOVA, non-normal with rank-sum test, and categorical variables with chi-square test.

Characteristics of participants were described and compared according to the prevalence of DN. According to the results, amino acids with significant differences in DN prevalence were screened out for further analysis. Spearman correlation was performed on the correlation between amino acids and DR. A binary logistic regression model stratified by DR condition was then used to obtain odds ratios (OR) for different amino acids to DN and their 95% confidence intervals (95%CI). Traditional risk factors for DN in T2D patients were adjusted by a structural adjustment program: model adjusted age, gender, BMI, SBP, DBP, TG, TC, HbA1c, HDL-C, LDL-C, Duration of DN, UA, and Crea. Then, the additive interaction between different drugs and DR was analyzed and the correlation coefficient of additive interaction was calculated. All analyses were performed using R version 4.1.0.

## 3. Result

### 3.1. Description of Study Subjects

Table 1 summarizes the select characteristics of the DN group and the non-DN group stratified by DR condition in the total population. The study included 1031 participants, with a mean age of 57.24 years old (SD: 13.82) and a mean BMI of 25.29 (SD: 3.85). Additionally, 548 patients were male (53.15%). 

Then, we divided the total population into two groups based on prevalence of DR. In the DR group, the mean age was 57.77 years old (SD: 9.96) and the mean BMI was 25.09 (SD: 3.31). Additionally, there were 73 males (45.1%) in this group. When compared according to the outcome of DN, we found that differences in TC, UA, and use of three oral antidiabetic drugs were statistically significant between the two groups. Patients with DN had higher TC and UA, and a greater proportion of using all three common drugs. In the T2D population without DR, the mean age was 57.14 years old (SD: 14.43) and the mean BMI was 25.33 (SD: 3.95). Additionally, there were 475 males without DR (54.7%). In this group, age, BMI, SBP, HDL-C, UA, Crea, and the use of Acarbose was statistically different between the second level groups divided by DN. The patients in the DN group were older, and have higher BMI, SBP, HDL-C, UA, and Crea. A larger percentage of people with DN use Acarbose. 

### 3.2. Differences in Individual Amino Acids According to the Appearance of DN

It is observed that the 10 amino acids of Leu, Phe, Trp, Tyr, His, Val, Gly, Thr, Cit, and Ser had significant differences in T2D patients with different DN prevalence (Table 2). Except for Cit, the concentrations of the other 9 amino acids in DN patients were lower than those in T2D patients.

### 3.3. Correlations between Amino Acids and DR and the Impacts of DR on Amino Acids for DN

We did correlation analysis on the selected 10 amino acids and DR (Figure 3). The results showed that except Cit, the remaining amino acids were positively correlated with DR, and the correlations were all statistically significant. Among amino acids, the correlation between Leu and Val was the strongest, reaching a strong correlation level (r = 0.84). Due to the significant correlation between amino acids and DR, we next analyzed the relationship between amino acids and DN risk stratified by the prevalence of DR (Table 3). The results showed that the protective effect of His, Trp, Val, and Thr on the risk of DN was no longer significant in the DR group. 

### 3.4. Addictive Interaction between Oral Hypoglycemic Drugs and DR

Table 4 shows the additive interactions between different drugs and DR. Concomitant use of Acarbose and Metformin increased the risk of DN (OR: 1.61, 95%CI: 1.13–2.29), and, although either Acarbose or Metformin alone increased the risk of DN, the risk of concomitant use of both drugs was higher than that of single use any one. Similar results can be observed in the additive interaction results of Acarbose and Sulfonylureas. In the interaction analysis of Sulfonylureas, Metformin, and DR, the highest risk of DN was using Sulfonylureas only in the presence of DR (OR: 2.95, 95%CI: 1.5–5.81). Followed by DR and the use of both Sulfonylureas and Metformin (OR: 2.56, 95%CI: 1.56–4.21).

## 4. Sensitive Analysis

After random forest imputation of missing values (UA = 187, TG = 288, TC = 289, Crea = 147), the effects of special amino acids for the risk of DN stratified by DR in T2D remained stable and significant in multi-variable analyses (Table 5). 

## 5. Discussion

In recent years, many studies have found that the risk of DN in DR patients is higher than that in the general T2D population. A study has found a significant association between DR and subsequent increased risk of DN in T2D patients, with younger patients at greater risk than older patients [20]. A study in Sudanese shows a significant association between DR and DN in adults with diabetes [21]. Additionally, there is a study which found that DR contributes to the diagnosis of DN in patients with T2D and kidney disease, but its severity may not be parallel to the presence of DN [4]. Klein et al. and Kofoed-Enevoldsen et al. suggested that the occurrence of DN and DR may be regulated by similar molecular pathways, which means that patients with DN may have already developed DR and patients with DR are vulnerable to develop DN. However, results in T2D have been inconsistent [22,23]. Excepting for population heterogeneity, the duration of T2D in different study populations is also an important influencing factor for the inconsistent results. At the same time, because DN can cause proteinuria, which shows significant renal lesions in patients, the inclusion of some patients with glomerulopathy who have the same results but do not have DN will also affect the results [24].

Our results showed that the protective effects of His, Trp, Val, and Thr on the risk of DN were affected by the prevalence of DR. A metabolomic study of DR found Asn, dimethylamine, His, Thr, and Gln to be the most variable metabolites in DR patients [25]. Another study confirmed that the metabolism of Gly, Ser Trp, and Thr is significantly disturbed in DR patients, especially the Trp metabolism [26]. Additionally, it is found that valine–leucine–isoleucine biosynthesis was also significantly disturbed in DR patients [27].

Some published studies have shown that Metformin can protect the kidney. However, the result obtained in our study shows that Metformin increases the risk of DN. We believe that excepting for population heterogeneity, the reason for the different results is that the use of Metformin leads to changes in the metabolism of other nutrients. Multiple studies have shown that long-term use of Metformin reduces Vitamin B_12_ (VB_12_) levels in the body [28,29,30]. A study in a North Indian population found that VB_12_ supplementation prevented the development of DN and improved the overall management of people with diabetes [31]. In animal experiments, it was found that both folic acid and VB_12_ can reduce the 24 h urinary albumin, and the combined effect is better [32]. High concentrations of homocysteine (Hcy) have been identified as a risk factor for DN [33,34]. Vitamin B supplementation can effectively reduce Hcy levels, of which VB_12_ is more effective in reducing Hcy concentration [35].

The study found that, although Acarbose can significantly improve blood sugar levels in DN patients, proteinuria did not improve [36]. Additionally, in mice experiments, Acarbose was found that it does not significantly reduce the incidence and severity of glomerulosclerosis [37]. This is consistent with our findings. Using Acarbose alone had no significant effect on the development of DN. However, the results showed that the effect of Metformin on DN was amplified when Metformin and Acarbose were used together. Additionally, our study showed that the effect of Sulfonylureas on DN was not significant when used alone. In published studies, some scholars have pointed out that gliquidone can improve the antioxidant response and delaying renal interstitial fibrosis by inhibiting the Notch/Snail1 signaling pathway, thereby improving the symptoms of DN. Gliquidone, as one Sulfonylureas, can ameliorate the diabetic symptoms of DN through inhibiting Notch / Snail1 signaling pathway, improving anti -oxidative response, and delaying renal interstitial fibrosis [38]. On the other hand, studies have also found that Glibenclamide (another kind of Sulfonylureas) should be used cautiously in patients with stages 2 and 3 DN. Additionally, Sulfonylureas are contraindicated in patients with stage 4 DN [39]. We believe that the difference may be caused by the different duration and stage of the DN in the study.

Based on the current research on microvascular disease, our study further explored the effect of amino acid metabolism changes on the risk of DN in different DR conditions. At present, there are few published studies on the relationship between amino acid metabolism and DN. The amino acids selected from our results can provide certain directions and ideas for further refinement of DN metabolism research. At the same time, our research has some shortcomings. (1) Due to the nature of the cross-sectional study, we could not prove the causality between DR and progression of DN [40,41], and the order of occurrence between DR and DN cannot be determined. At present, there are also some studies that believe the occurrence of DN promotes the development of DR. For example, a study conducted in Pakistan indicated that DN is an independent risk factor for the development and progression of DR [42]. However, unlike DN, the DR is not only a kind of microvascular disease, but can also cause a certain degree of nerve damage. Due to its more complex pathogenic factors and more related lesions, it is believed that its onset time has been studied more extensively before DN. More prospective studies are needed to prove the causal relationship between them; (2) The lack of available vitamin indicators makes it difficult to verify the effects of drugs on vitamins. (3) Other than the three drugs mentioned in the article, the effects of other oral hypoglycemic agents were not included in the study. Our laboratory team is conducting monitoring of vitamin indicators and improving the information of more types of hypoglycemic drugs to subsequently validate our results.

In conclusion, we selected amino acids that have protective effect on the risk of DN, and found that DR patients had a higher risk of developing DN. Additionally, the use of oral hypoglycemic drugs can also increase the risk of DN, with combining use of drugs has worse effect than that of any one drug alone. The occurrence of the disease is a complex process with multiple effects, and more follow-up studies are needed to confirm the association between different risk factors, so as to better intervene in the disease. 

## Figures and Tables

**Figure 1 ijerph-20-04623-f001:**
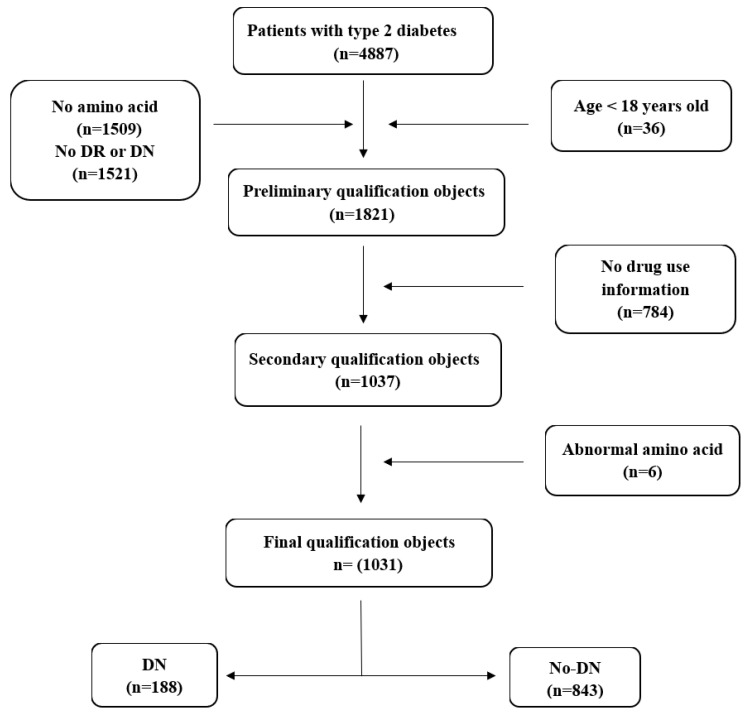
Schematic diagram of subject screening process. Three screening were made in the total population. A total of 1821 patients were screened out after the preliminary exclusion of patients with deletion of amino acids, age, and the prevalence of DR and DN. Then, 1037 patients were screened out through the second screening according to the use of three drugs. Additionally, 1031 patients were screened out through the exclusion of amino acid outliers, including 188 patients with DN and 843 patients with type 2 diabetes. DN, diabetic nephropathy, DR, diabetic retinopathy.

**Figure 2 ijerph-20-04623-f002:**
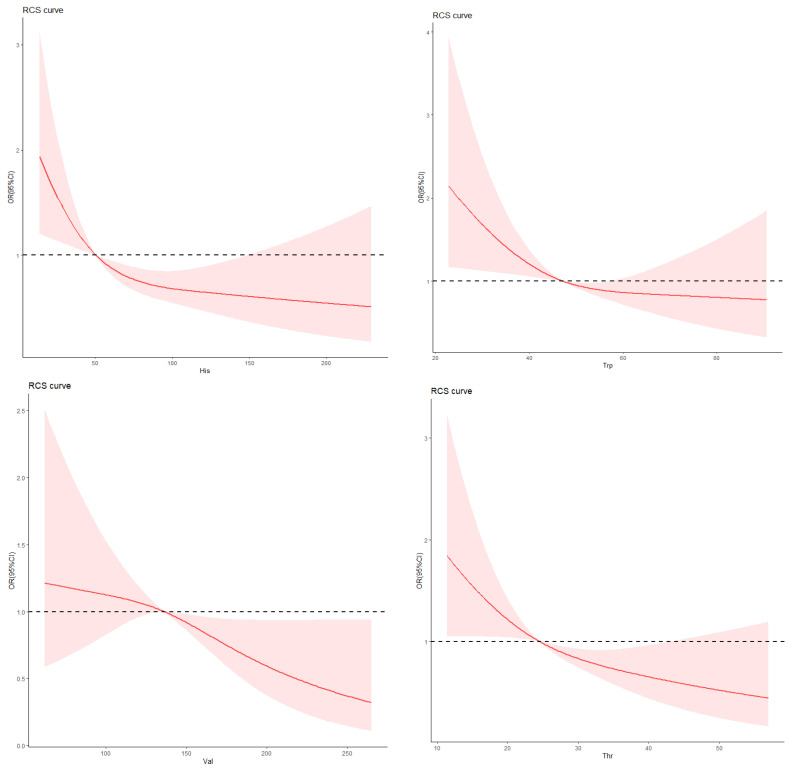
The RCS curve of His, Trp, Val and Thr. The red curve was derived from multivariate analysis that adjusted for age, gender, body mass index, systolic blood pressure, diastolic blood pressure, low-density lipoprotein cholesterol, high-density lipoprotein cholesterol, triglyceride, glycosylated hemoglobin, uric acid, serum creatinine, and duration of diabetic nephropathy. His, histidine; Trp, tryptophan, Val, valine; Thr, threonine.

**Figure 3 ijerph-20-04623-f003:**
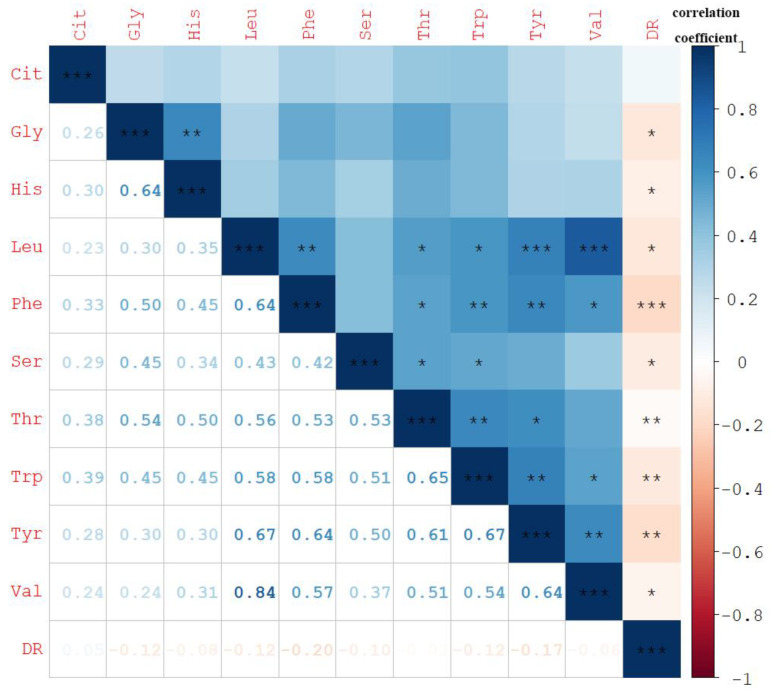
Spearman correlation coefficients between selected plasma amino acids and DR (n = 1031). The number and the color of grid represent the magnitude of the correlation coefficient and the correlation, respectively. Darker blue means more positive correlation, and darker red means more negative correlation. DR was divided into binary variables according to whether or not. Leu, leucine; Phe, phenylalanine; Trp, tryptophan; Tyr, tyrosine; Val, valine; Gly, glycine; Thr, threonine; Cit, citrulline; His, histidine; Ser, serine, DR, diabetic retinopathy. * *p*-value < 0.05, ** *p*-value < 0.01, *** *p*-value < 0.001.

**Table 1 ijerph-20-04623-t001:** Clinical and biochemical characteristics of participants according to the occurrence of DN.

Variables	Total People	DR				Non-DR			
	Mean/Number (SD or %)	Total DR	Non-DN	DN	P^α^	Total Non-DR	Non-DN	DN	P^α^
		Mean/Number (SD or %)	Mean/Number (SD or %)	Mean/Number (SD or %)		Mean/Number (SD or %)	Mean/Number (SD or %)	Mean/Number (SD or %)	
Age (years)	57.24 ± 13.82	57.77 ± 9.96	57.99 ± 10.27	57.44 ± 9.55	0.731	57.14 ± 14.43	56.67 ± 14.53	59.96 ± 13.55	0.019
Male sex	548 (53.15)	73 (45.1)	42 (42.9)	31 (48.4)	0.592	475 (54.7)	410 (55.0)	65 (52.4)	0.657
Weight (kg)	70.34 ± 13.18	68.89 ± 12.09	68.70 ± 12.29	69.17 ± 11.86	0.808	70.61 ± 13.36	70.30 ± 13.37	72.49 ± 13.19	0.091
Height (cm)	167.00 [160.00, 172.00]	164.00 (160.00, 172.00)	163.00 (158.25, 172.00)	165.00 (160.00, 171.25)	0.369	167.00 (160.00, 173.00)	168.00 (160.00, 173.00)	167.00 (160.00, 172.00)	0.963
BMI (kg/m²)	25.29 ± 3.85	25.09 ± 3.31	25.10 ± 3.36	25.08 ± 3.27	0.981	25.33 ± 3.95	25.22 ± 3.90	26.02 ± 4.16	0.035
SBP (mmHg)	140.39 ± 23.99	145.60 ± 25.26	144.18 ± 22.84	147.77 ± 28.62	0.379	139.42 ± 23.63	138.33 ± 23.44	145.94 ± 23.83	0.001
DBP (mmHg)	82.43 ± 13.50	83.04 ± 13.44	82.89 ± 13.01	83.27 ± 14.16	0.862	82.32 ± 13.51	82.29 ± 13.51	82.45 ± 13.58	0.904
HbA1c (%)	9.30 (7.70, 11.00)	9.25 (7.70, 10.90)	8.90 (7.70, 10.67)	9.75 (7.77, 11.35)	0.155	9.30 (7.70, 11.00)	9.30 (7.70, 11.00)	9.20 (7.60, 10.93)	0.682
Triglyceride (mmol/L)	1.69 (1.13, 2.39)	1.69 (1.18, 2.40)	1.71 (1.18, 2.68)	1.67 (1.20, 2.22)	0.766	1.69 (1.12, 2.39)	1.69 (1.11, 2.39)	1.70 (1.14, 2.41)	0.648
TC (mmol/L)	4.64 (3.86, 5.29)	4.81 (4.07, 5.59)	4.70 (4.00, 5.33)	4.98 (4.23, 6.25)	0.048	4.61 (3.83, 5.25)	4.61 (3.82, 5.21)	4.61 (3.83, 5.34)	0.633
HDL-C (mmol/L)	1.02 (0.85, 1.25)	1.04 (0.88, 1.29)	1.04 (0.89, 1.23)	1.04 (0.87, 1.38)	0.517	1.01 (0.85, 1.25)	1.00 (0.84, 1.24)	1.07 (0.92, 1.28)	0.020
LDL-C (mmol/L)	2.78 (2.19, 3.36)	2.87 (2.34, 3.45)	2.85 (2.34, 3.30)	2.89 (2.31, 3.76)	0.438	2.77 (2.15, 3.34)	2.76 (2.16, 3.33)	2.79 (2.12, 3.38)	0.430
UA	311.00 (245.95, 381.50)	314.50 (251.25, 373.02)	303.00 (250.00, 357.00)	341.10 (273.75, 391.50)	0.027	310.00 (244.00, 383.00	306.70 (242.00, 378.00)	334.00 (254.75, 395.50)	0.030
Crea	58.97 (49.02, 73.30)	56.81 (48.76, 71.37)	56.81 (46.68, 67.84)	57.35 (50.39, 81.98)	0.223	59.49 (49.09, 73.55)	59.00 (49.15, 71.14)	65.08 (48.87, 89.05)	0.005
Duration of DM	5 (0, 10)	13 (6, 20)	14.5 (6, 20)	11.5 (5, 18.63)	0.598	4 (0, 10)	3 (0, 10)	8 (1, 12)	0.027
Use of Metformin	358 (34.7)	51 (31.5)	23 (23.5)	28 (43.8)	0.011	307 (35.3)	263 (35.3)	44 (35.5)	0.999
Use of Acarbose	364 (35.3)	53 (32.7)	24 (24.5)	29 (45.3)	0.010	311 (35.8)	255 (34.2)	56 (45.2)	0.024
Use of Sulfonylureas	146 (14.2)	29 (17.9)	12 (12.2)	17 (26.6)	0.034	117 (13.5)	98 (13.2)	19 (15.3)	0.608

BMI, body mass index. SBP, systolic blood pressure. DBP, diastolic blood pressure; HbA1c, glycated hemoglobin; TG, triglyceride; HDL-C, high-density lipoprotein cholesterol; LDL-C, low-density lipoprotein cholesterol; UA, uric acid; Crea, creatinine, DM, diabetes mellitus, DN, diabetic nephropathy, DR, diabetic retinopathy. Data are mean ± standard deviation, median (IQR), or n (%). *p* values were derived from the *t*-test for normally distributed variables, Mann-Whitney U test for skewed distributions, Chi-square test (or fisher test if appropriate) for categorical variables. *p* < 0.05 was defined as statistically significant. P^α^ is obtained by comparing the two groups divided by the prevalence of DN.

**Table 2 ijerph-20-04623-t002:** Plasma amino acids levels according to DN condition.

	DN	NDN	*p*
Ala, µmol/L	122.11 (96.07, 145.55)	126.06 (99.68, 154.88)	0.223
Asn, µmol/L	72.02 (61.29, 88.20)	75.24 (62.00, 89.81)	0.143
Leu, µmol/L	122.36 (93.65, 149,63)	127.09 (102.11, 158.23)	0.013
Phe, µmol/L	42.96 (34.73, 52.11)	45.83 (37.66, 55.54)	0.014
Lys, µmol/L	120.43 (83.61, 169.10)	128.62 (94.60, 176.37)	0.125
Glu, µmol/L	98.61 (81.76, 118.35)	97.83 (81.18, 121.61)	0.685
Trp, µmol/L	44.47 (37.23, 53.40)	48.26 (38.91, 57.06)	0.007
Tyr, µmol/L	42.95 (33.37, 52.36)	46.40 (37.52, 56.81)	<0.001
His, µmol/L	44.61 (33.18, 68.59)	52.50 (36.01, 80.28)	0.006
Val, µmol/L	129.61 (107.95, 155.44)	137.34 (114.52, 162.08)	0.017
Pip, µmol/L	130.63 (95.81, 176.48)	126.63 (94.68, 173.57)	0.690
Arg, µmol/L	10.62 (5.71, 16.25)	10.52 (5.94, 17.38)	0.427
Gly, µmol/L	191.63 (150.83, 243.37)	204.94 (152.85, 269.09)	0.037
Pro, µmol/L	465.84 (347.03, 602.25)	435.97 (335.14, 585.28)	0.284
Thr, µmol/L	23.22 (18.22, 28.95)	24.78 (19.96, 30.57)	0.005
Cit, µmol/L	21.95 (16.60, 27.43)	19.43 (15.16, 25.35)	0.001
Gln, µmol/L	6.70 (4.91, 9.14)	6.94 (5.07, 9.28)	0.555
Met, µmol/L	16.61 (13.66, 21.23)	17.20 (14.46, 21.17)	0.112
Ser, µmol/L	49.60 (42.15, 59.64)	52.13 (43.40, 64.91)	0.048
Orn, µmol/L	17.49 (13.28, 22.46)	17.39 (12.91, 23.89)	0.828
Asp, µmol/L	28.10 (21.41, 35.32)	28.40 (20.90, 37.87)	0.914
Pip, µmol/L	130.63 (95.81, 176.48)	126.63 (94.68, 173.57)	0.690
Cys, µmol/L	1.21 (0.87, 1.69)	1.28 (0.95, 1.67)	0.195

DN, diabetic nephropathy; Ala, alanine; Asn, asparagine; Leu, leucine; Phe, phenylalanine; Trp, tryptophan; Tyr, tyrosine; Val, valine; Arg, arginine; Gly, glycine; Pro, proline; Thr, threonine; Cit, citrulline; Gln, glutamine; His, histidine; Lys, lysine; Met, methionine; Ser, serine; Orn, ornithine; Glu, glutamate; Asp, aspartate; Pip, piperamide, Cys, cysteine. Data are mean ± standard deviation, median (IQR), or n (%).

**Table 3 ijerph-20-04623-t003:** Odds ratio of significant amino acid for the risk of diabetic nephropathy stratified by diabetic retinopathy.

	DR		No-DR	
	OR (95%CI)	*p*	OR (95%CI)	*p*
Univariable Model				
His	0.92 (0.67, 1.27)	0.619	0.79 (0.63, 1)	0.036
	0.73 (0.38, 1.4)	0.339	0.74 (0.5, 1.08)	0.117
Trp	1.15 (0.84, 1.58)	0.386	0.79 (0.65, 0.97)	0.022
	0.85 (0.45, 1.62)	0.631	0.63 (0.43, 0.92)	0.018
Val	1.17 (0.85, 1.61)	0.329	0.74 (0.6, 0.91)	0.004
	1.18 (0.63, 2.21)	0.611	0.62 (0.42, 0.91)	0.014
Thr	0.9 (0.65, 1.24)	0.513	0.7 (0.55, 0.9)	0.002
	0.92 (0.49, 1.73)	0.799	0.64 (0.43, 0.93)	0.021
Multivariable Model				
His	0.98 (0.68, 1.39)	0.892	0.69 (0.52, 0.9)	0.003
	0.73 (0.35, 1.52)	0.401	0.56 (0.37, 0.86)	0.007
Trp	1.21 (0.83, 1.74)	0.323	0.74 (0.59, 0.93)	0.007
	0.96 (0.46, 2)	0.92	0.51 (0.33, 0.77)	0.001
Val	1.27 (0.86, 1.87)	0.231	0.72 (0.57, 0.92)	0.007
	1.35 (0.65, 2.79)	0.416	0.59 (0.39, 0.91)	0.016
Thr	0.97 (0.68, 1.39)	0.864	0.64 (0.48, 0.85)	<0.001
	1.02 (0.49, 2.11)	0.961	0.55 (0.36, 0.84)	0.005

Each amino acid corresponds to the first behavior of a numerical variable result, the second behavior of a dichotomic variable result. Trp, tryptophan; Val, valine; Thr, threonine; His, histidine; DR, diabetic retinopathy. Multivariable Model was adjusted for age, gender, body mass index, systolic blood pressure, diastolic blood pressure, low-density lipoprotein cholesterol, high-density lipoprotein cholesterol, triglyceride, total cholesterol, glycosylated hemoglobin, uric acid, creatinine, and duration of DN.

**Table 4 ijerph-20-04623-t004:** Interactive effects between different oral hypoglycemic drugs and DR for DN risk.

	OR (95% CI)	*p* Value
DR vs. No-DR	3.79 (2.56, 5.62)	<0.001
Metformin vs. No-Metformin	1.51 (1.05, 2.17)	0.027
Acarbose vs. No-Acarbose	1.87 (1.32, 2.66)	<0.001
Sulfonylureas vs. No-Sulfonylureas	1.81 (1.16, 2.81)	0.011
Additive interaction model of Acarbose and Metformin		
No-Acarbose and No-Metformin	Reference	
No-Acarbose and Metformin	1.09 (0.63, 1.88)	0.766
Acarbose and No-Metformin	1.26 (0.84, 1.89)	0.267
Acarbose and Metformin	1.61 (1.13, 2.29)	0.008
RERI	0.449 (0.064, 0.834)	
APAB	0.273 (0.113, 0.434)	
S	3.31 (0.800, 13.697)	
Additive interaction model of Acarbose and Sulfonylureas		
No-Acarbose and No-Sulfonylureas	Reference	
No-Acarbose and Sulfonylureas	1.65 (0.84, 3.24)	0.157
Acarbose and No-Sulfonylureas	1.74 (1.21, 2.51)	0.003
Acarbose and Sulfonylureas	1.92 (1.36, 2.72)	<0.001
RERI	0.426 (0.062, 0.790)	
APAB	0.270 (0.110, 0.431)	
S	3.832 (0.563, 26.106)	
Additive interaction model of Sulfonylureas, Metformin and DR		
No-Sulfonylureas and No-Metformin and No-DR	Reference	
No-Sulfonylureas and No-Metformin and DR	2.39 (1.69, 3.39)	<0.001
No-Sulfonylureas and Metformin and No-DR	1.79 (1.26, 2.55)	0.001
No-Sulfonylureas and Metformin and DR	2.18 (1.52, 3.13)	<0.001
Sulfonylureas and No-Metformin and No-DR	1.62 (0.69, 3.79)	0.282
Sulfonylureas and No-Metformin and DR	2.95 (1.5, 5.81)	0.003
Sulfonylureas and Metformin and No-DR	2.11 (1.23, 3.61)	0.008
Sulfonylureas and Metformin and DR	2.56 (1.56, 4.21)	<0.001
RERI	0.405 (0.028, 0.782)	
APAB	0.259 (0.088, 0.430)	
S	3.578 (0.549, 23.306)	

DR, diabetic retinopathy; DN, diabetic nephropathy. RERI risk due to interaction, AP attributable proportion due to interaction, S synergy index. Significant elative excess risk due to two of interaction (RERI) > 0, attributable proportion due to interaction (AP) > 0 or synergy index (S) > 1 indicates a significant additive interaction. Multivariable Model was adjusted for age, gender, body mass index, systolic blood pressure, diastolic blood pressure, triglyceride, total, cholesterol, low-density lipoprotein cholesterol, high-density lipoprotein cholesterol, glycosylated hemoglobin, uric acid, creatinine, and duration of DN.

**Table 5 ijerph-20-04623-t005:** Odds ratio of significant amino acid for the risk of diabetic nephropathy stratified by diabetic retinopathy after random forest imputation of missing values.

	DR		No-DR	
	OR (95%CI)	*p*	OR (95%CI)	*p*
Univariable Model				
His	0.92 (0.67, 1.27)	0.619	0.79 (0.63, 1)	0.036
	0.73 (0.38, 1.4)	0.339	0.74 (0.5, 1.08)	0.117
Trp	1.15 (0.84, 1.58)	0.386	0.79 (0.65, 0.97)	0.022
	0.85 (0.45, 1.62)	0.631	0.63 (0.43, 0.92)	0.018
Val	1.17 (0.85, 1.61)	0.329	0.74 (0.6, 0.91)	0.004
	1.18 (0.63, 2.21)	0.611	0.62 (0.42, 0.91)	0.014
Thr	0.9 (0.65, 1.24)	0.513	0.7 (0.55, 0.9)	0.002
	0.92 (0.49, 1.73)	0.799	0.64 (0.43, 0.93)	0.021
Multivariable Model				
His	0.95 (0.66, 1.36)	0.783	0.69 (0.53, 0.91)	0.004
	0.71 (0.34, 1.48)	0.356	0.57 (0.37, 0.86)	0.007
Trp	1.22 (0.84, 1.78)	0.293	0.73 (0.58, 0.92)	0.005
	0.99 (0.48, 2.06)	0.975	0.5 (0.33, 0.76)	0.001
Val	1.21 (0.82, 1.77)	0.339	0.72 (0.56, 0.91)	0.005
	1.28 (0.62, 2.65)	0.508	0.58 (0.38, 0.89)	0.012
Thr	0.96 (0.68, 1.36)	0.823	0.64 (0.48, 0.84)	<0.001
	1.06 (0.51, 2.21)	0.872	0.54 (0.35, 0.82)	0.003

Each amino acid corresponds to the first behavior of a numerical variable result, the second behavior of a type of variable result. Trp, tryptophan; Val, valine; Thr, threonine; His, histidine; DR, diabetic retinopathy. Multivariable Model was adjusted for age, gender, body mass index, systolic blood pressure, diastolic blood pressure, low-density lipoprotein cholesterol, high-density lipoprotein cholesterol, triglyceride, total cholesterol, glycosylated hemoglobin, uric acid, creatinine, and duration of DN.

## Data Availability

The datasets for this manuscript are not publicly available at the time of publication. Requests to access the datasets should be directed to fangzhongze@tmu.edu.cn.

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
