# Peer review of "Prevalence of Diabetic Retinopathy and Use of Common Oral Hypoglycemic Agents Increase the Risk of Diabetic Nephropathy—A Cross-Sectional Study in Patients with Type 2 Diabetes"

_ijerph, 2023, doi:10.3390/ijerph20054623_

Round 1
Reviewer 1 Report
Please add the objectives of the study
The duration of diabetes can be mentioned
Was the hypoglycaemic agent the first line of treatment?
Since how long patient been on this treatment?
These questions need to be addressed
Reviewer 2 Report
Dear Authors,
This is a retrospective study evaluating relevant differences in amino acid concentration between patients with type 2 diabetes and chronic renal disease. Moreover, the effect of certain medications (such as metformin, sulfonylureas, and acarbose) on DR risk was calculated. The paper contains much information, but it needs to be better organized, and possible methodological complaints are also evident.
1. Title does not represent the study contents, as it did not include some relevant information, such as the type of study and its aims.
2. The introduction should also contain information about the role of metabolomics in chronic diabetes-related complications.
3. Methods should include more information about the diagnostic criteria of chronic complications and better define the selection of participants.
4. Methodology: some concerns could be addressed when interpreting the results expressed in figure 2. DR was a dichotomic variable (y or n), or it was classified according to an ordinal scale (i.e., DR severity?). Ensure that the statistical test (Spearman) is appropriate for the specific purpose.
5. When calculating the ORs of DR based on amino acid profile, which cut-off did you use to dichotomize the value? Please express them clearly in the paper (methods or results).
6. According to table 1, patients were on metformin (35%), acarbose (35%), and sulfonylureas (14%), with a total of 84% of treated patients. Did the other 16% of patients not assume any medications?
7. Background chronic complications are expected to affect the kind of prescribed anti-hyperglycemic medications. For example, the frequency of metformin in patients with chronic renal diseases could be lower compared to that observed in individuals without renal complaints. This generates an inevitable selection bias due to the application of general recommendations and guidelines (and it is undoubtedly correct). As the cross-sectional study design, therefore, the calculation of ORs attributable to the use of specific medications it is not informative about the risk.
